# Traffic Noise and Mental Health: A Systematic Review and Meta-Analysis

**DOI:** 10.3390/ijerph17176175

**Published:** 2020-08-25

**Authors:** Janice Hegewald, Melanie Schubert, Alice Freiberg, Karla Romero Starke, Franziska Augustin, Steffi G. Riedel-Heller, Hajo Zeeb, Andreas Seidler

**Affiliations:** 1Institute and Policlinic of Occupational and Social Medicine, Faculty of Medicine, Technische Universität Dresden, 01307 Dresden, Germany; janice.hegewald@tu-dresden.de (J.H.); melanie.schubert@tu-dresden.de (M.S.); alice.freiberg@tu-dresden.de (A.F.); karla.romero_starke@tu-dresden.de (K.R.S.); franzi_augustin@web.de (F.A.); 2Institute of Sociology, Faculty of Behavioral and Social Sciences, Chemnitz University of Technology, Thüringer Weg 9, 09126 Chemnitz, Germany; 3Institute for Social Medicine, Occupational Medicine and Public Health, University of Leipzig, 04103 Leipzig, Germany; Steffi.Riedel-Heller@medizin.uni-leipzig.de; 4Department of Prevention and Evaluation, Leibniz-Institute for Prevention Research and Epidemiology—BIPS, 28359 Bremen, Germany; zeeb@bips.uni-bremen.de; 5Health Sciences, University of Bremen, 28359 Bremen, Germany

**Keywords:** noise, transportation, traffic noise, noise pollution, road traffic noise, aircraft noise, railway noise, anxiety, depression, disruptive behavior disorders, psychology, cognition disorders

## Abstract

Recent evidence suggests that traffic noise may negatively impact mental health. However, existing systematic reviews provide an incomplete overview of the effects of all traffic noise sources on mental health. We conducted a systematic literature search and summarized the evidence for road, railway, or aircraft noise-related risks of depression, anxiety, cognitive decline, and dementia among adults. We included 31 studies (26 on depression and/or anxiety disorders, 5 on dementia). The meta-analysis of five aircraft noise studies found that depression risk increased significantly by 12% per 10 dB L_DEN_ (Effect Size = 1.12, 95% CI 1.02–1.23). The meta-analyses of road (11 studies) and railway traffic noise (3 studies) indicated 2–3% (not statistically significant) increases in depression risk per 10 dB L_DEN_. Results for road traffic noise related anxiety were similar. We did not find enough studies to meta-analyze anxiety and railway or aircraft noise, and dementia/ cognitive impairment and any traffic noise. In conclusion, aircraft noise exposure increases the risk for depression. Otherwise, we did not detect statistically significant risk increases due to road and railway traffic noise or for anxiety. More research on the association of cognitive disorders and traffic noise is required. Public policies to reduce environmental traffic noise might not only increase wellness (by reducing noise-induced annoyance), but might contribute to the prevention of depression and anxiety disorders.

## 1. Introduction

Anthropogenic environmental noise is pervasive in industrialized countries. According to the European Environmental Agency, approximately one in five people in the European Union was exposed to road traffic noise levels exceeding 55 dB L_DEN_ (yearly weighted day-evening-night noise average) in 2017 [1]. A growing body of evidence indicates that exposure to traffic noise may be detrimental to health. In 2018, the World Health Organization (WHO) released guidelines with recommendations for protecting health and for giving policy guidance in the European Region [2]. The WHO based their guidelines on systematic reviews commissioned to evaluate the health effects of traffic noise. One of these systematic reviews by Clark and Paunovic [3] examined studies of noise-related risks for various mental health outcomes published until 2015. This review did not include a meta-analysis and found no effect of road traffic noise on depression and anxiety. For aircraft noise, very low evidence of harmful effects was found for depression and anxiety measured using information on medication intake and interview measures. There were no studies for railway noise available. Another WHO review by Clark and Paunovic [4] examining associations between traffic noise and cognition found no studies on adult cognition. 

Clark et al. [5] updated both WHO reviews until March 2019. For depression/ anxiety, 24 newly published studies were included in the review, but again no meta-analysis was conducted. The review found low quality evidence for no effect of road traffic noise on the incidence of vascular dementia and very low quality evidence for a harmful effect of road traffic noise on emergency admissions or dementia symptoms. 

Recently, Dzhambov and Lercher [6] updated the WHO review for the effect of road traffic noise on depression and anxiety using a meta-analytic approach. The authors included 10 studies in their quantitative summarization with 15 estimates for depression and five for anxiety. In sum, they found an increased risk for depression (4%) and anxiety (12%) due to increasing road traffic noise/combination of environmental noise. However, these results failed to reach statistical significance (depression: odds ratio (OR) = 1.04; 95% confidence interval (CI) 0.97–1.11 and anxiety: OR = 1.12; 95% CI 0.96–1.30). So far, a quantitative summarization is missing for aircraft and railway noise. Without any quantitative summary of study results, it is difficult to understand how risks for mental disorders might be associated with traffic noise exposures. 

In addition, there are indications that not just traffic noise intensity but the source of traffic noise might also affect mental health. An emotional response to noise in the form of annoyance may be one way that noise contributes to mental health problems [7], and the WHO review on noise annoyance shows that noise annoyance varies greatly between traffic noise sources [8]. These reactions to noise could be due to the diverse characteristics of noise from each traffic source. Fluctuations in noise differ for each form of traffic noise, and the intermittency ratio (a metric of noise fluctuation) is positively associated with annoyance to railway and aircraft noise [9]. Traffic related sleep disturbances might also mediate the pathway between traffic noise exposure and mental health problems. Research demonstrates that each traffic noise source differentially affects subjective and objective sleep measures. Road and railway noise are more likely to cause an awakening than aircraft noise [10], while aircraft and railway noise negatively influence subjective sleep quality [11]. Our own study of traffic noise found aircraft traffic noise between >45–≤50 dB increased the odds of an incident depressive episode by 1.18 (95% CI 1.16–1.21) [12]. On the other hand, we observed a similar increase in odds for much higher road traffic noise of ≥70 dB (OR = 1.17; 95% CI 1.10–1.25). Therefore, it is important to differentiate between various sources of traffic noise when considering their associations with mental health problems. 

In this systematic review, we aimed to summarize the evidence regarding risks for mental health problems, in particular depression, anxiety, cognitive decline, and dementia among adults exposed to varying levels of road traffic, railway traffic, or aircraft noise by conducting a systematic review and meta-analysis of cohort, case-control, cross-sectional, and ecological studies. 

## 2. Materials and Methods 

### 2.1. Research Question and Study Eligibility

We sought published studies examining traffic noise (aircraft, road traffic, and railway noise) exposures and the risks of psychological complaints and diseases (further specification below). Our literature search included both children and adults. This paper focuses on the results for adult populations. Results for children were published recently [13]. While no protocol was published specifically for this research question, we followed the procedures outlined in a protocol for our systematic review on non-auditory health complaints and diseases due to aircraft noise registered on PROSPERO (CRD42013006004). This systematic review diverged from the registered study protocol by also considering exposures to road and railway traffic noise, and focusing on mental health complaints.

We further specified our research question and inclusion criteria according to Population-Exposure-Outcome characteristics. We included studies considering the general population (P), and excluded animal studies or studies on occupational populations (i.e., airport personnel or road or railway maintenance workers). Exposure (E) to road traffic, railway, and/or commercial aircraft noise should have been assessed objectively and quantified with actual measurements or noise models. We excluded studies of military aircraft noise, and did not include studies considering only neighborhood noise. We included studies of mental health and behavioral outcomes (O) corresponding to the diagnoses in chapter V of the ICD-10 (F00–F99), in particular depressive episodes (F32.–; F33.–), anxiety disorders (F40.–), as well as dementia in Alzheimer disease (F00.–), vascular and unspecified dementia (F01.–; F03.–), and mild cognitive disorders (F06.7). Outcomes should have been assessed with validated screening instruments, such as the General Health Questionnaire (GHQ), that can also identify undiagnosed problems and sub-clinical levels of mental health disorders, or based on diagnosed disorders, e.g., self-reported, routine data, prescribed medications specific to a psychological disorder (Table 1). 

We included primary studies using any forms of observational study design, i.e., ecological, cross-sectional, case-control, and cohort studies. Studies using an ecological or cross-sectional design typically do not contribute the same level of scientific evidence as prospective cohorts or even case-control studies; this is reflected in the assessments of methodological study quality. Reviews, editorials, and letters to the editors, as well any publication with incomplete information on study methods and results were excluded from the review. We included studies published in any language with an English or German title and abstract. Grey literature that did not appear in a peer-reviewed journal (e.g., conference proceedings, research reports) was included if sufficient information on study methods and results were provided. When information was lacking, we attempted to contact the publication authors.

### 2.2. Information Sources and Search 

We searched the electronic literature databases MEDLINE (Pubmed), Embase (Ovid), PsycINFO (ProQuest), and PSYNDEX (EBSCOhost) without any time limitations until December 11, 2019. Pubmed was searched using the search string: [(“depression” OR “affective” OR “anxiety” OR “panic” OR “dysthymia” OR “dementia” OR Alzheimer* OR “mental” OR psychi* OR psychol* OR “annoyance”) AND (“noise” AND (“aircraft” OR “airways” OR airplane* OR airline* OR “jet” OR “flight” OR rail* OR “train” OR “road” OR “highway” OR “street” OR “traffic” OR “transport”)]. This search string was adapted for the other databases accordingly.

Reviews and references of included studies were searched for additional references. We also searched the proceedings of the German Society for Acoustic (*Deutsche Gesellschaft für Akustik*, DAGA) and the INTER-NOISE conferences for studies. We did not search any grey literature database. We used no geographic or language restrictions though a German or English title and abstract was an inclusion criteria.

### 2.3. Study Selection and Data Collection 

The search results were imported into an Endnote reference management system database and duplicate references removed at import. The titles and abstracts were screened independently by two authors (M.S. and J.H.) for inclusion and exclusion. Disagreements regarding inclusion were discussed, and often included in the full-text screening to err on the side of caution. The full-texts of articles were screened by two independent reviewers (K.R.S., F.A., A.F., J.H., A.S., or M.S.) and disagreements resolved in meetings. 

We extracted the following information: study design,region,study population size,population characteristics (age and sex distributions),population sampling information (recruitment times, response and follow-up),outcomes considered and how they were assessed (instruments used),noise exposure sources considered,noise assessment, including the noise levels considered, andstudy results.

Further details, such as adjustment for confounders, conflicts of interest, and funding sources were also extracted in the comments section of our extraction form. Extraction of data was done by one reviewer (F.A., K.R.S., or M.S.) and checked for accuracy by a second reviewer (A.F. or J.H.). Either fully-adjusted results or the results of what the authors described their “main analyses” were extracted and included in the meta-analysis.

At least two reviewers (K.R.S., F.A., A.F., J.H., A.S., or M.S.) assessed the methodological quality by using a hybrid tool comprising characteristics of both the SIGN (Scottish Intercollegiate Guidelines Network 2004) and CASP (Critical Appraisal Skills Programme 2004/2006) assessment tools used previously for other reviews of noise-related health effects [14]. For example, the tool includes questions on appropriate recruitment of study population, measurement of exposure and outcome. In addition, it is evaluated if all important confounding factors (in our study age, sex and education/socioeconomic state) have been considered and taken into account in the design and analysis. 

A study was given the quality rating of ‘++’, ‘+’, or ‘–‘. The study quality ratings are also included under the comments section of the detailed extraction form (Appendix A). Divergent assessments of quality were discussed in meetings and resolved collectively.

### 2.4. Meta-Analysis

Meta-analyses were conducted with Stata 14 [15] to obtain a pooled risk estimates per 10 dB if at least three studies considering the same exposure source and outcome were available. The following procedure was used to pool the risk estimates:If risk estimates were reported for categories of noise, study-specific risk estimates per 10 dB linear increase of traffic noise were estimated by applying the generalized least-squares model for trend estimation of summarized dose-response data using the *glst* Stata package [16]. The generalized least squares for trend estimation takes into account the fact that the risk estimates from a single study do not fulfill the assumption of independence required of weighted linear regression. A generalized least-squares model was estimated for each applicable study separately as a fixed-effect model using the logarithm of the risk estimates as the dependent variable and average of the noise level categories (with the reference group set to zero) as the independent variable. If necessary, noise exposure values were converted to day-evening-night weighted-24 h means (L_DEN_) according to Brink et al. [17] prior to the modeling (see Table 2).The reported and self-calculated risk estimates per 10 dB Lden were pooled using a DerSimonian and Laird random-effects meta-analysis. The random-effects model was chosen because heterogeneity between study populations can be expected. This method weights each effect estimate by the inverse of its (within-study) variation and the heterogeneity between studies (between-study variation). The Stata package *metan* was used to conduct the random-effects meta-analysis and to create forest plots [18].

Heterogeneity was quantified by calculating I^2^ according to the following formula: I^2^ = (Q − df/Q) × 100, where *Q* is the chi-squared statistic and *df* is the degrees of freedom (number of studies − 1) [19,20]. The Q chi-squared statistic is the weighted sum of the differences between each study’s effect estimate and the pooled estimate squared. 

In sensitivity analyses, we excluded studies with low methodological quality when at least three studies with a ‘+’ or ‘++’ rating was available. This was done to determine the possible direction and impact of study bias on the pooled results. We also assessed the influence of each included study on the pooled results by excluding it from the meta-analysis, in a so-called ‘leave-one-out analysis’. Where five or more studies were available, we also visually considered publication bias using a funnel plot. 

## 3. Results

### 3.1. Study Selection

We found a total of 5428 citations in electronic databases, and after duplicates were removed 4252 citations remained for the title and abstract screening. The full-text screening included 224 articles found in the electronic databases and an additional 57 articles (*n* = 281) found from other sources, such as the reference lists of included studies or review articles [3,5,6,21,22,23,24,25,26,27,28,29,30,31,32,33,34,35,36] and the INTERNOISE and the German Acoustical Society conference proceedings. We excluded a total of 236 full-text articles from further consideration. Our reasons for excluding studies are summarized in the PRISMA flow diagram (Figure 1). We were unable to retrieve the full-texts of nine studies. The references of excluded studies and the reasons for exclusion are listed in Appendix A. After the full-text screening, we identified and extracted 31 articles describing 28 studies considering the influence of traffic noise on the mental health of adults (20 studies on depression, 11 on anxiety, and 5 on dementia/ cognitive decline).

### 3.2. Depression

We included 20 publications that considered the effect of traffic noise on depression [12,37,38,39,40,41,42,43,44,45,46,47,48,49,50,51,52,53,54,55,56]. An abbreviated summary of study characteristics is shown in Table 3, and the complete tables are in Appendix A.

Studies mostly included males and females in the study population with the exception of Stansfeld et al. [49,50] who only included males (based on the *Caerphilly Collaborative Heart Disease Study*) and He et al. [57] who considered the hospital records of women with one pregnancy during the study period. 

Regarding the study regions, most studies were conducted in Europe: five studies were from the Netherlands [39,43,45,54,55] and another five were from Germany [12,41,47,48,53]. Two studies each were performed in Finland [42,46] and the UK [49,50,51]. Further studies were conducted in Norway [52] and France [37], and another international study considered several European countries [38]. The two studies conducted outside of Europe were from Canada [56] and Japan [44].

Nine of the included studies used a cross-sectional design [37,38,42,43,45,46,48,52,53] and five studies had a case-control design [12,39,41,44,55]. Another three were cohort studies [47,49,50,56]. One of the included studies had a prospective intervention design (natural experiment) [51]. Generaal et al. [40] performed a pooled analysis of eight Dutch studies (*NEMESIS-2*, *HELIUS*, *NTR*, *NESDA*, *HOORN*, *LASA*, *NL-SH*, *Generations^2^*) using a cross-sectional design.

Most of the included studies assessed depression using validated questionnaires, such as various versions of the General Health Questionnaire (GHQ) [37,44,49,50,51,54,58], the Hopkins Symptom Checklist (HSCL-25) that assesses psychological stress (depression/ anxiety) [52], and the Clinical Interview Schedule-Revised (CIS-R) survey tool [50,51]. Another study recorded depressive moods with the Patient Health Questionnaire (PHQ)-9 [43]. Tzivian et al. also used the General Depression Scale (Center for Epidemiological Studies Depression Scale-CES-D) scale to record depressive symptoms in older subjects. Generaal et al. [40] recorded depression and anxiety disorders according to the Diagnostic and Statistical Manual for Mental Disorders (DSM)-IV criteria using the Composite International Diagnostic Interview (CIDI). Three studies considered antidepressant use together with the GHQ-12 [54], the CES-D [47], or with depression/ anxiety recorded with the Kessler psychological distress scale (K10) [45]. Three other studies analyzed the use of antidepressants in the last two weeks [38] or during the survey year [42,46]. Schreckenberg et al. [48] recorded the consumption of mood-influencing drugs and sedatives in the past 12 months. Three studies used ICD-9 or ICD-10 diagnoses from routine health insurance data [12,41] or hospital records [57]. Zock et al. [55] used the International Classification of Primary Care (ICPC) code. 

Altogether, twelve studies assessed the effect of road traffic noise on depression [12,38,42,43,45,46,47,49,50,51,52,53,55]. Furthermore, seven studies investigated the association between aircraft noise and depression [12,37,38,41,44,48,54]. The influence of railway noise on depression was studied in three publications [12,45,55]. Generaal and colleagues [39,40] used a combined measure of road, aircraft, and railway traffic. He et al. [56] used a land-use regression model of measured noise that included several noise sources such as road traffic noise and aircraft and railway traffic in the vicinity to airports and railways, respectively.

#### 3.2.1. Synthesis of Results

In the meta-analysis, we included 15 studies that used an indication of a clinical depression, such as diagnosed depressive episodes, anti-depressive use or depressive symptoms detected with a screening tool. Different sources of traffic noise were examined in separate sub-groups. Eleven of the studies were included in the meta-analysis for road traffic noise [12,38,42,43,45,47,49,50,52,53,55,60], five studies were included in the meta-analysis of aircraft noise [12,37,38,48,54], and three studies were included in the meta-analysis that considered the effect of railway traffic noise on depression [12,45,55]. The pooled risk estimates for depression per 10 dB L_DEN_ road and railway noise were marginally increased but not statistically significant (Figure 2). Moderate to substantial heterogeneity was observed for the road traffic noise results (I^2^ = 60.2%, 11 studies) and considerable heterogeneity was observed for railway traffic noise results (I^2^ = 95.6%, 3 studies). A 12% increase in depression risk was observed per 10 dB L_DEN_ aircraft noise that was also statistically significant (ES 1.12; 95% CI 1.02–1.23; 5 studies). Heterogeneity for the aircraft noise results was negligible (I_2_ = 0%), in part due to the wide confidence intervals of the individual effect estimates. 

#### 3.2.2. Risk of Bias across Studies

Five studies on road traffic noise and depression received an acceptable methodological quality rating of ‘+’ [12,45,51,53,55], and the remaining studies received a low rating (’−’). When we considered only the studies with an acceptable methodological quality rating (‘+’), the pooled effect size increased slightly (ES 1.04; 95% CI 0.99–1.10; I^2^ = 79.7%). While the Stansfeld et al. [49,50] study was technically rated (‘+’) for its longitudinal analysis, we derived unadjusted risk estimates for the meta-analysis from reported distributions that may be subject to bias. Excluding these results from the analysis lowered the results somewhat (ES 1.02; 95% CI 0.99–1.06; I^2^ = 62.8%).

For aircraft noise, only the Seidler et al. [12] study had an acceptable methodological quality (rating ‘+’), and all three studies investigating the effect of railway noise on depression were of acceptable quality (rating ‘+’). Therefore, no sensitivity analysis of study quality was done for these noise sources.

As part of the “leave-one-out” analysis, each study was omitted from meta-analysis in a stepwise fashion. This resulted in pooled effect sizes ranging from 1.02 to 1.04 for road traffic noise, thus indicating high stability of pooled results even if studies with more weight were left out. In the subgroup of aircraft noise studies, removing the larger Seidler et al. [12] study from the meta-analysis reduced the ES so that it was no longer significant (ES 1.06; 95% CI 0.95–1.19). The removal of the other studies had nearly no effect on the meta-analysis results. Results are shown in the Appendix A. Funnel plots for depression and road traffic (Appendix A) and aircraft noise (Appendix A) also show some indications of publication bias (asymmetry).

### 3.3. Anxiety Disorders

Eleven studies were included that considered the effect of traffic noise on anxiety disorders [38,39,41,42,45,49,50,55,61,62,63]. An abbreviated summary of study characteristics is shown in Table 4, and the complete tables are in Appendix A. 

With the exception of the *Caerphilly Collaborative Heart Disease Study* that only included males [49,50], the study populations consisted of males and females. Ten of the studies were conducted in Europe. Three were conducted in the Netherlands [39,45,55], one in France [61], Finland [46], Germany [41], Finland [59], Norway [63], and the UK [49,50]. The study by Floud et al. [38] was an international cooperation between six European countries. A further study came from Canada [62]. A cross-sectional design was used in six studies [38,42,45,46,62,63], three studies used a cohort design [50,55,61], and two studies used a case-control design [39,41]. 

A validated instrument for diagnosing anxiety disorders was used by four studies: CIDI: [39], State-Trait Anxiety Inventory (STAI): [62,63], GHQ-30: [49,50]. Four studies used prescription and/or health insurance data [41,45,55,61], and another three studies considered the self-reported use of anxiolytics within a specific time window [38,42,46]. 

Nine studies investigated the association between road traffic noise and anxiety [38,42,45,46,49,50,55,61,62,63]. Two studies each considered the effect of aircraft noise [38,41] and railway noise [45,55]. One study used a combined measure of road, aircraft, and railway traffic noise [39].

#### 3.3.1. Synthesis of Results for the Meta-Analysis

Studies were included in the meta-analysis if a screening tool with cut-point or other indication of an anxiety disorder was used. Too few studies examined aircraft (n = 1) and railway traffic noise (n = 2) to conduct a meta-analysis. Six studies considered the risk of anxiety due to road traffic noise and were pooled. The meta-analysis of anxiety and road traffic noise studies resulted in an effect size of 1.02 (95% CI 0.98–1.06). The heterogeneity of the meta-analysis was moderate at I2 = 61.0% (Figure 3).

#### 3.3.2. Risk of Bias across Studies

Of the road traffic noise studies, only Klompmaker et al. [45] and Zock et al. [55] received an acceptable quality rating ‘+’, so no sensitivity analysis to examine the potential effect of bias on the results was conducted. Both of these studies were also the only to also investigate the effect of railway noise on anxiety disorders. The one aircraft noise study (Figure 3) received a low-quality rating ‘–‘ [38].

The “leave-one-out” analysis showed a lower risk for road-traffic noise related anxiety disorders when the Klompmaker et al. [45] study was excluded (ES = 1.00, 95% CI 0.98–1.02). The Bocquier et al. [61] study reported results stratified by neighborhood deprivation levels, and found an increased risk of anxiety due to road traffic noise among people in the low deprivation group. Since these effect estimates represent separate populations (no single person is counted twice), estimates for each of the three deprivation groups were included in the meta-analysis. Removing all three subpopulations from the analysis increased the pooled ES to 1.07 (95% CI 1.01–1.13–1.08) and the pooled estimate reached statistical significance (Appendix A). The funnel plot for anxiety and road traffic noise is included in the online supplementary (Appendix A).

### 3.4. Dementia and Alzheimer’s Disease

We included five studies investigating road traffic noise related effects on dementia and cognitive impairment [64,65,66,67,68]. A brief summary of the studies is provided in Table 5. Fuks and colleagues [67] included only females in the study. Both sexes were investigated in all other studies. Two studies were from Germany [67,68]. One study each was conducted in Sweden [64], the UK [65], and Spain [66]. No study investigated the effect of railway traffic or aircraft noise on the cognitive function of adults. 

Two studies examined dementia diagnosed by a doctor (specialty unknown) or a specialist [64,65]. Furthermore, mild cognitive impairment (MCI) and cognitive function was assessed with test batteries in the *Heinz Nixdorf Recall Study* [68] and the SALIA study [67], respectively. Linares and colleagues [66] used dementia-related emergency hospital admissions. 

There were some indications for an effect of road traffic noise on cognition. Tzivian and colleagues [68] observed a significant increase in risk of cognitive impairment for road traffic noise levels of 60 dB L_DEN_ or more (OR = 1.40; 95% CI 1.03–1.91) and L_NIGHT_ levels of 55 dB or more (OR = 1.80; 95% CI 1.07–3.04). Fuks et al. [67] found an influence of road traffic noise on the cognitive function (total) for L_DEN_, but not for L_NIGHT_. In addition, the Spanish study by Linares et al. [66] showed an association between dementia-related hospital admissions and noise levels during the day. However, the studies on dementia [64,65] found no evidence for a road-traffic related dementia risk.

#### Synthesis of Results 

Generally, four studies met the inclusion criteria for a meta-analysis [64,65,67,68]. However, the outcomes differed among studies. Whereas two of the studies considered diagnosed dementia [64,65], a further study investigated mild cognitive impairment [68], and Fuks et al. considered lower than expected cognition test scores based on the German age- and educational level standards for women [67]. Thus, we did not perform a meta-analysis.

## 4. Discussion

Based on the meta-analysis of five studies, we found evidence that aircraft noise increased depression risks by 12% per 10 dB L_DEN_ of (ES = 1.12, 95% CI 1.02–1.23). The meta-analyses of road and railway traffic noise indicated 3% and 2% increases in depression risk per 10 dB L_DEN_, respectively, but these results were not statistically significant. The results for road traffic noise and anxiety were similar to depression (2% increases). However, we did not find enough studies to conduct meta-analyses for the risk of anxiety due to railway traffic or aircraft noise and generally for traffic noise related dementia.

Dzhambov and Lercher [6] obtained comparable results for road traffic noise on depression risks in their meta-analysis. They report a pooled risk for depression of OR = 1.04 (95% CI: 0.97–1.11) per 10 dB L_DEN_ road traffic noise with a moderate heterogeneity. However, the selection of studies differed somewhat between our systematic reviews. Dzhambov and Lercher [6] included the eight risk estimates from the “pooled” analysis of cohort studies conducted by Generaal et al. 2019, as well as an effect estimate from the He et al. 2019 [56] study. We excluded both because traffic noise sources were not considered separately. In contrast, we included five additional studies on depression risks for road traffic noise in our meta-analysis [42,43,49,50,52,58].

For anxiety and road traffic noise, we found the pooled risk estimate increased by 2% per 10 dB L_DEN_ road traffic noise, but was not statistically significant (ES = 1.02, 95% CI 0.98–1.06). In contrast, Dzhambov and Lercher [6] report a higher, but also not statistically significant, risk increase of 12% (OR = 1.12, 95% CI 0.96–1.30) per 10 dB L_DEN_ for anxiety disorders. Again, the studies included in our meta-analysis differed from the studies included in the Dzhambov and Lercher [6] review. Dzhambov and Lercher [6] included a study by Generaal et al. [39] which used a combined measure of road traffic, aircraft, and railway noise, and observed a very high risk for anxiety disorders (OR =1.86, 95% CI 1.31–2.73 per 10 dB L_DEN_). This study inflated the heterogeneity of their study results. According to our own re-calculation of their results, excluding this risk estimate lowers the pooled estimate and heterogeneity considerably (ES = 1.10, 95% CI 1.05–1.15). However, our additional inclusion of the French retrospective cohort by Bocquier et al. [61] which considered prescriptions of anxiolytics, benzodiazepine derivatives, and analogues in relation to road traffic noise, and the Finnish prospective cohort of workers by Halonen et al. [42], which examined anxiolytic prescriptions with road traffic noise, resulted in a lower pooled estimate. When the study by Bocquier and colleagues [61] was excluded, we observed a significant increase for an anxiety disorder of 7% per 10 dB road traffic noise (95% CI 1.01–1.13). In this study, the number of prescriptions was used to define an anxiety disorder. However, as these medications are also prescribed for other indications, this may be an insufficient proxy for anxiety disorders.

Clark and colleagues [3,5] considered anxiety and depression together in their evaluation, but differentiated on how anxiety and depression were assessed (i.e., medication use, interview or self-reported measures). In the first review commissioned by the WHO, Clark et al. [3] found a total of four studies examining the noise-related risks for depression and anxiety medication use (one aircraft, three road studies) providing a very low quality of evidence. Their review also identified four studies (road) with self-reported depression, anxiety, and psychological symptoms as the outcome, and two studies (one aircraft, one road) considering depressive and anxiety disorders measured with interviews. The quality of evidence for these outcomes was also rated as very low. In summary, Clark and Paunovic [3] found harmful effects for aircraft noise based on medication intake and interview-assessed depression and anxiety (very low quality evidence), but no effect of road traffic noise on anxiety and depression. No studies were available for railway noise. In the review update, Clark and colleagues [5] found low quality evidence for a harmful effect of aircraft and road traffic noise on depression and anxiety measured with interviews (two aircraft, four road studies) as opposed to very low quality evidence in their WHO review. Furthermore, very low quality evidence for harmful effects was found for depression and anxiety medication use (two road studies). There was a very low quality evidence for no effect of road traffic noise on self-reported depression, anxiety, and psychological symptoms (seven road studies).

Three of the five studies on road traffic noise-related cognitive disorders that we identified found some indications for an increased risk of dementia. Clark and Paunovic [4] observed no effect of road traffic noise on the incidence of vascular dementia based on two studies (low quality). Very low-quality evidence for a harmful effect of road traffic noise was suggested for dementia-related emergency admissions (two studies) and cognitive assessment of dementia symptoms (one study). Recent evidence suggests that exposures to the air pollutants nitrogen dioxide (NO_2_), nitrous oxides (NO_X_), carbon monoxide (CO), and particulate matter ≤2.5 μm (PM_2.5_) are risk factors for cognitive decline and dementia [70]. However, the impact of road traffic noise on cognition is still unclear. Studies indicate air pollution and road traffic noise are correlated. In particular, moderate (0.40) to high correlations (0.84) have been described for NO_2_ and NO_X_ with road traffic noise [29,71,72,73,74]. Thus, harmful effects on cognition might be due to air pollutants rather than due to road traffic noise *per se*. Future research on cognitive function should incorporate both risk factors in their study design and clarify the isolated effects of road traffic noise and air pollution on cognitive health.

Limitations of this systematic review are that we included different measures of depression and anxiety. Most researchers considered the relationship between transportation noise and mental disorders by using validated questionnaires or interviews. However, some research considered how prescriptions or self-reported use of medication is associated with noise exposure. We classified anti-depressants use/prescriptions as having a depressive disorder, and the use/prescriptions of anxiolytics, as well as benzodiazepine derivatives, and analogues were defined as having an anxiety disorder. However, these medications are prescribed for a range of mental disorders that are also often comorbid [75,76]. Anti-depressants, such as selective serotonin re-uptake inhibitors, are frequently prescribed for patients with generalized anxiety disorder (GAD) [77,78]. Benzodiazepine derivatives and analogues are frequently prescribed to patients with a major depressive disorder. For example, Liu et al. [79] showed in a retrospective analysis of a large claims data set that 33% of patients diagnosed with a major depressive disorder were prescribed benzodiazepines and 6% were treated with anxiolytics. In addition, having a mental disorder may not necessarily correspond to medication intake. For generalized anxiety disorder (GAD), it has been shown that one in five patients receive only psychotherapy, 31% only drug treatment, and 27% both. In addition, 23% of GAD-patients receive no treatment at all [80]. Moreover, psychotropic medications use may also occur without having a mental disorder [80]. Thus, prescription/ medication use data may not be a valid indicator for the presence of a specific mental disease.

Furthermore, we also included studies using health claims data. Substantial proportions of persons suffering from a mental disorder do not seek treatment, or only seek treatment after long delays. According to the WHO World Mental Health Survey Initiative, on average, only one in five persons make treatment contact within a year of mental disorder onset in Europe [81]. Here, the lowest rates were observed for Germany (14%) and highest rates for the Netherlands (28%). Other population-based studies made comparable observations. For instance, approximately only one in five study participants in the European study of the Epidemiology of Mental Disorders with a 12-month prevalence of a mental disorder reported having used health services in the previous year [82]. Seeking mental health care is strongly related to age, with older adults showing a lower perceived need for treatment [81,83]. Males are also less likely to seek health care [82]. Moreover, the type of mental disorder may also affect health care use. It has been shown that individuals with mood disorders are more likely to use health care services (37%) than those suffering from an anxiety disorder (26%) [82]. Thus, health claims data might be subject to bias, as the prevalence of mental disorders might be underestimated. However, studies looking at health claims data over a longer period of time are more likely to detect serious cases of mental disorders. These large datasets are also a good source of objective measures since they use standardized international coding and data are not specifically collected for research purposes.

In this systematic review, we identified indications of an association between transportation noise and mental health problems, with the most convincing evidence of a relationship found between aircraft noise exposure and increased risk of depression. However, most of the included studies in this systematic review were of low methodological quality. There is a need for high quality prospective population-based studies of railway noise, aircraft noise, and studies considering traffic noise exposure and dementia. All studies on traffic noise suffer from some degree of misclassification bias due to inexact assessments of noise exposure that, at best, are modeled to one spot on the outside of a home. Without information on how long study participants actually spend at home, there is likely to be both over- and underestimation of the true exposure. The common convention of using the loudest house façade may also systematically overestimate noise exposure, and better traffic noise measurements will improve the precision of future studies [84]. Improved noise measurements in future studies might also help attenuate the problem of heterogeneity, which was considerable in the meta-analyses for road and railway traffic noise. However, methodological diversity usually occurs in meta-analysis. Therefore, it is argued that heterogeneity is inevitable [20].

In order to support the development of public policies that effectively protect mental health, it is helpful to understand the pathomechanism of this relationship. Babisch [85] suggests that transportation noise could impact health directly by stimulating areas of the brain such as the reticular activating system, or indirectly by eliciting an emotional or cognitive response to perceived noise. Possible biological pathways discussed in the literature also include increased noise-related sleep deprivation, annoyance, and stress reactions [8,86]. For instance, Sygna and colleagues [52] showed that effects of road traffic noise on mental health were only present in subjects with poor sleep quality. A study by Halonen et al. 2012 [87] found that persons with higher trait anxiety had higher insomnia symptoms at L_NIGHT_ > 50 dB. Moreover, noise annoyance may also play an important role, as studies suggest that noise annoyance affects the association between reported sleep quality and transportation noise [88]. An annoyance-mediated pathway could also explain greater risk of depression due to aircraft noise that we observed compared to other types of traffic noise at the similar noise levels. Aircraft noise causes more annoyance than other transportation noise sources at comparable levels of L_DEN_ [8]. This also raises the question of the adequacy of average noise levels for representing the traffic noise-related risk of depression, as this might not adequately describe other potentially relevant characteristics of noise such as fluctuations and maximum noise levels. For example, in the NORAH (NOise-Related Annoyance, cognition, and Health) study of health-risks, it was shown that individuals with low (<40 dB) average levels of aircraft noise but with higher nightly maximum levels over 50 dB have an increased depression risk [12]. Future research should therefore also consider the importance of maximum noise levels and also of other metrics such as the intermittency ratio to investigate health risks [89]. Understanding how traffic-related noise impacts mental-health can help to shape targeted interventions, but as annoyance is also part of mental health in a broader sense, a general noise reduction can benefit mental health.

## 5. Conclusions

The effect of transportation noise on the risk of mental health disorder has been subject to many discussions. Our review adds to three recent systematic reviews [4,5,6], by considering each source of traffic noise separately using meta-analyses. Although our review differed somewhat from these previous reviews, overall our observations also show that aircraft noise exposure increases the risk for depression (/anxiety). Our results also suggest that road traffic noise may exert comparatively small effects on depression and anxiety risks; however, few studies had an acceptable methodological quality and the study design (i.e., outcome and exposure assessment) varied greatly. We found too few studies to make conclusions on the effects of railway noise on mental health. We encourage future research to design high quality (prospective) studies assessing all traffic noise sources in the same population. In the meantime, public policies that reduce environmental traffic noise can help to prevent depression and anxiety disorders.

## Figures and Tables

**Figure 1 ijerph-17-06175-f001:**
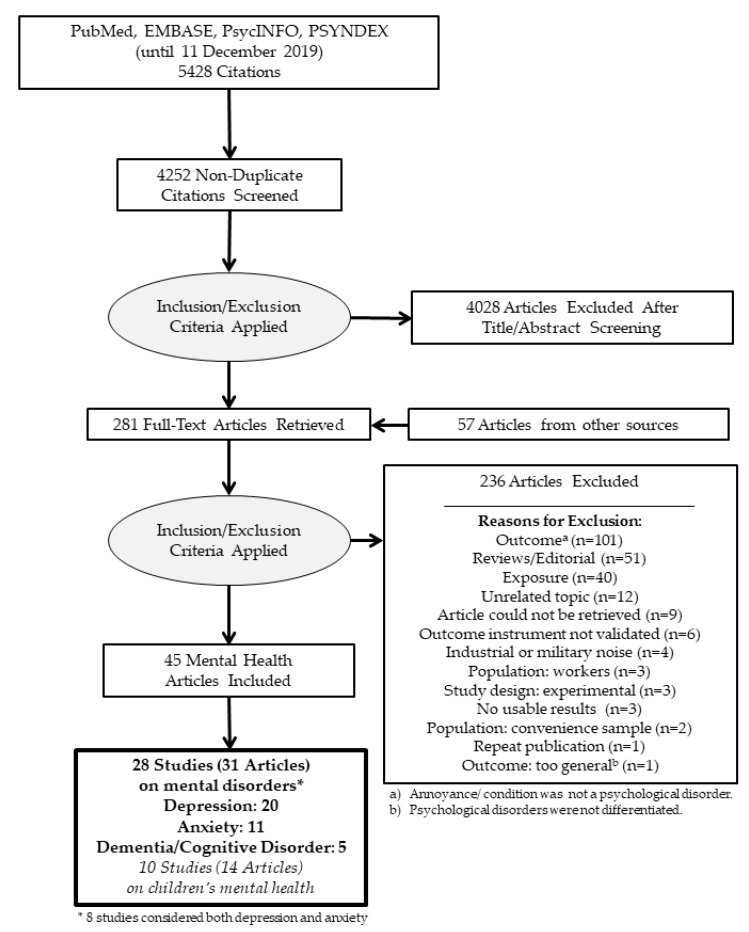
PRISMA flow diagram.

**Figure 2 ijerph-17-06175-f002:**
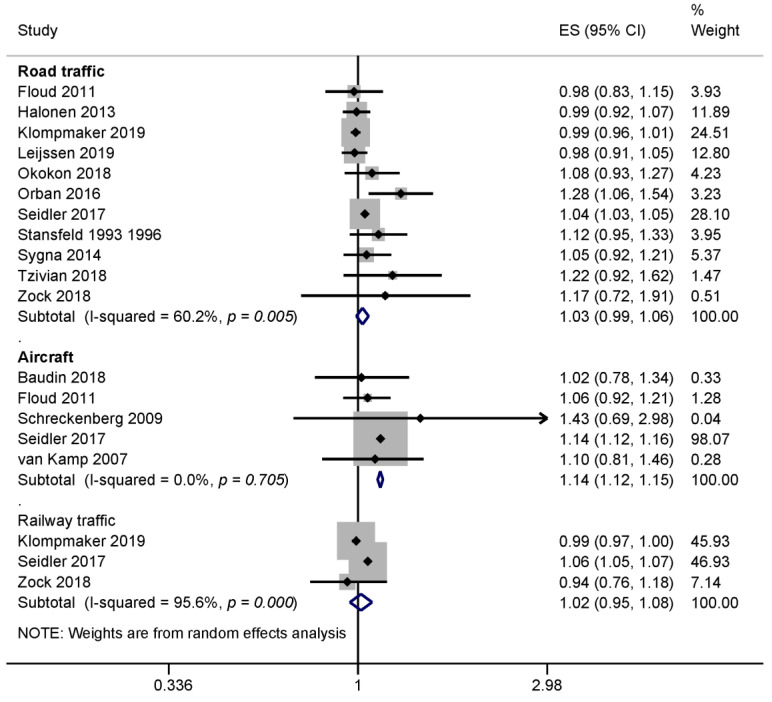
Forest plots of various noise sources and the estimates of effect size (ES) for depression (based on anti-depressant use, depressive episodes diagnoses, detected with validated a screening instrument). All ES were converted to represent the risk per 10 dB L_DEN_.

**Figure 3 ijerph-17-06175-f003:**
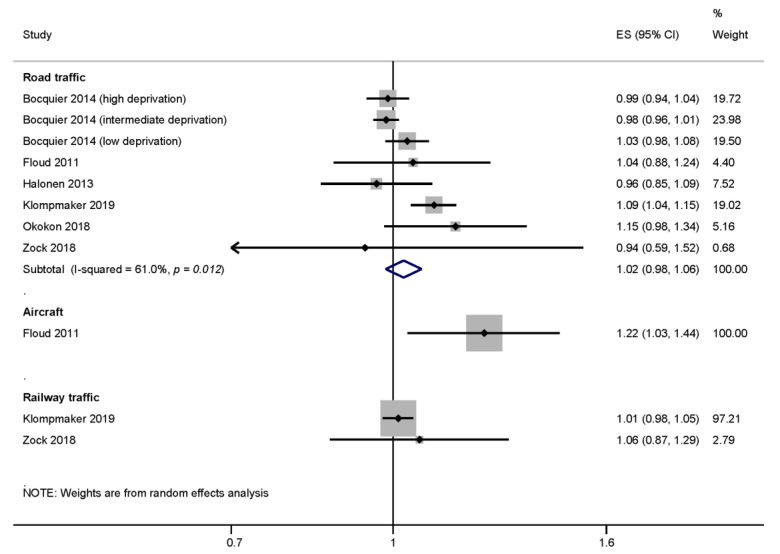
Forest plots of various noise sources and the estimates of effect size (ES) for anxiety (based on anxiolytics use, diagnoses, detected with a validated screening instrument). All ES were converted to represent the risk per 10 dB L_DEN_.

**Table 1 ijerph-17-06175-t001:** Study inclusion and exclusion criteria.

	Inclusion	Exclusion
**Population**	Adults sampled from the general population	Animal studies; occupational populations, non-representative (i.e., convenience) samples
**Exposure**	Road traffic, railway, or aircraft noise assessed objectively (i.e., measurements or noise modelling at place of residence)	Military aircraft noise; studies considering only neighborhood noise
**Outcomes**	Psychological complaints and disorders [in particular: dementia, vascular or Alzheimer, mild cognitive disorder (F00–F03, F06.7), depressive episodes (F32,–; F33,–) and anxiety disorders (F40,–)] (i.e., diagnosed disorders, e.g., self-reported, routine data; prescribed medications specific to a psychological disorder; validated screening instrument)	Annoyance; sleep disturbance; conditions not directly related to a clinical diagnosis; screening instrument was not validated

**Table 2 ijerph-17-06175-t002:** Converting to L_DEN_ according to Brink et al. [17].

Converting	Road Traffic Noise	Railway Traffic Noise	Aircraft Noise
**L_eq24h_ to L_DEN_**	Leq_24h_ + 3.3 dB	Leq_24h_ + 6.1 dB	Leq_24h_ + 3.6 dB
**L_D_ to L_DEN_**	L_D_ + 2.0 dB L_eq,16h_ + 1.5 dB	L_D_ + 6.0 dB L_eq,16h_ + 5.9 dB	L_D_ + 2.1 dB L_eq,16h_ + 2.3 dB
**L_DN_ to L_DEN_**	L_DN_ + 0.7 dB	L_DN_ + 0.4 dB	L_DN_ + 1.1 dB
**L_N_ to L_DEN_**	L_N_ + 8.0 dB	L_N_ + 6.6 dB	L_N_ + 9.9 dB

**Table 3 ijerph-17-06175-t003:** Depression.

Study	Region(s) Study Design Quality Score (++,+,-)	Population	Outcome Assessment	Noise Source	Noise Measures	Noise Categories	Effect Estimates; Meta-Analysis (Yes, No with Reasons)
Baudin et al. 2018 [37]	France cross-sectional (-)	*n* = 1244 (M = 549, F = 695) 18–≥75 Years	GHQ-12	Aircraft	L_DEN_ L_eq,24h_ L_D, 6::00 a.m.-10:00_ _p.m._ L_N, (not specified)_	<40 50–54 55–59 ≥60 dB	Odds Ratio (adjusted) yes
Floud et al. 2011 [38]	UK, NL, Sweden, Italy, Germany, Greece, cross-sectional (-)	*n* = 4861 (M = 2404, F = 2457) 45–70 Years	Anti-depressive use (last 2 weeks, self-reported)	Aircraft road	L_eq, 7:00 a.m.-11:00 p.m._ or L_eq, 6::00 a.m.-10:00 p.m._ L_N, 11:00 p.m.-7:00 a.m._ or L_N, 10:00 p.m.-6:00 a.m._	L_Aeq,16h_ ^a^ 35–76 dB (aircraft) ^a^ 45–76 dB (road) L_N_ ^a^ 30–70 dB (aircraft) ^a^ 45–70 dB (road)	Odds Ratio (adjusted) yes
Generaal et al. 2019a [39]	NL case-control/cross-sectional (-)	*n* = 2980 (M = 1007, F = 1973) 18–65 years	Recent diagnosis (major depression and dysthymia) based on CIDI ^b^	Aircraft Road rail (combined)	L_DEN_	-	Odds Ratio (adjusted) no, exposure sources were not considered separately
Generaal et al. 2019b [40]	NL pooled analysis (-)	*n* = 32,487 ≥18 years 8 studies included	CIDI, PHQ-9, HADS-D, DES-D, 4DSQ, BDI-II	Aircraft Road rail (combined)	L_DEN_	-	Odds Ratio (adjusted) no, exposure sources were not considered separately
Greiser et al. 2010 [41]	Germany case-control (-)	*n* = 511,742 with depression: *n* = 3136 (M = 981, F = 2155) >39 years	Routine health insurance data ^c^	aircraft	L_N, 10:00 p.m.-6:00 a.m._ L_eq, 6:00 a.m.-10:00 p.m._ L_eq, 11:00 p.m.-1:00 a.m._ L_eq, 3:00 a.m.-5:00 a.m._ L_eq,24h_	L_eq,24h_ ≥35 dB otherwise ≥40 dB	Odds Ratio (adjusted) no, main noise effects not reported without interaction terms
He et al. 2019 [56]	Canada cohort (-)	F = 140,456	Hospital records ^d^	outdoor noise (mostly road)	L_eq,24h_ L_DEN_ L_N, 11:00 p.m.-7:00 a.m._	<55 dB 55.0–59.9 dB 60.0–64.9 dB ≥65 dB	Hazard Ratio (adjusted) no, exposure sources were not considered separately
Halonen et al. 2013, 2014 [42,59]	Finland cross-sectional (-)	*n* = 15,611 (M = 3086, F = 12,525) 21–76 years	Anti-depressive use ^e^	road	L_DEN_	≤45 dB 45.1–50 dB 50.1–55 dB 55.1–60 dB >60 dB	Odds Ratio (adjusted) yes
Leijssen et al. 2019 [43]	NL cross-sectional (-)	*n* = 23,293 (M = 9920, F = 13,373) 18–70 years	PHQ-9	road	L_eq,24h_	45–54 dB 55–59 dB 60–64 dB 65–69 dB ≥70 dB	Odds Ratio (adjusted) yes
Klompmaker et al. 2019 [45]	NL cross-sectional (+)	*n* = 354,827 (M = 161,045, F = 193,782) ≥19 years	Prescriptions of antidepressants (ATC code ^e^), K10	road railway	L_DEN_	-	Odds Ratio (adjusted) yes
Miyakawa et al. 2007 [44]	Japan cross-sectional (-)	*n* = 188 (M = 101, F = 87)	GHQ-28	aircraft	L_DEN_	55–59 dB 59–65 dB	Odds Ratio (adjusted) no, noise was not measured for ‘unexposed’ sample
Okokon et al. 2018 [46]	Finland cross-sectional (-)	*n* = 5860 (M = 2497, F = 3363) 55.0 years	Anti-depressive use (a) 1 week, (b) 1–4 weeks, (c) 1–12 months, (d) over 1 year ago	road	L_DEN_	≤45 dB, 45.1–50 dB, 50.1–55 dB, 55.1–60 dB ≥60 dB	Odds Ratio (adjusted) yes
Orban et al. 2016 [47]	Germany cohort (+)	*n* = 3300 (M = 1715, F = 1585) 45–74 years	CES-D Anti-depressive use (last week)	road	L_DEN_ L_N, 10:00 p.m.-6:00 a.m._	L_DEN_ >55 vs. ≤55 dB; ≤55 dB >55–≤60 dB >60–≤65 dB >65 dB L_N_ >50 dB	Relative Risks (adjusted) yes
Schreckenberg et al. 2009 [48]	Germany cross-sectional (-)	*n* = 2311 (M = 1034, F = 1276) <18–≥80 years	Daily use of mood-controlling, psychotropic drugs (last 12 months)	aircraft	L_eq, 6:00 a.m.-10:00 p.m._ L_N, 10:00 p.m.-6:00 a.m._ NAT_55,6:00 a.m.-10:00 p.m._ ^f^ NAT_55, 10:00 p.m.-6:00 a.m._ L_max55, (6:00 a.m.-10:00 p.m.)_ ^f^ L_max55, (10:00 p.m.-6:00 a.m.)_	40–45 dB 45–50 dB 50–55 dB 55–60 dB 60–65 dB For L_eq,22-06h_: reference < 40 dB	Odds Ratio (adjusted) yes
Seidler et al. 2017 [12]	Germany case-control (+)	*n* = 1,026,670 (77,295 cases, 578,246 controls) ≥40 years	Routine health insurance data ^g^	Aircraft Road railway	L_eq,24h._ L_N, 10:00 p.m.-6:00 a.m._ (further measures considered for aircraft noise)	<40, max <50 dB <40, max ≥50 dB ≥40–<45 dB ≥50–<55 dB ≥55–<60 dB ≥60–<65 dB ≥65–<70 dB ≥70 dB; Continuous analysis ^h^	Odds Ratio (adjusted) yes
Stansfeld et al. 1993, 1996 [49,50]	UK cohort (+) cross-sectional (-)	*n* = 2398 (only men) 50–64 Years	GHQ-30	road	L_eq,6:00 a.m.-10 p.m._	51–55 dB 56–60 dB 61–65 dB 66–70 dB	Means/ Percent (adjusted) yes (*unadjusted RRs derived from distributions in 1993 paper*)
Stansfeld et al. 2009b [51]	UK intervention field study (-)	*n* = 387 households	CIS-R GHQ-28	road	L_eq,10:00 a.m.-5:00 p.m._ L10	*-*	Means/ Percent No *intervention field study*
Sygna et al. 2014 [52]	Norway cross-sectional (-)	*n* = 2898 (M = 1442, F = 1456) 18–˃78 years	HSCL-25	road	L_DEN_	*not reported*	Odds Ratio (adjusted) yes
Tzivian et al. 2018 [58]	Germany cross-sectional (analysis within cohort) (+)	*n* = 2050 50–80 years	CES-D	road	L_DEN_	Binomial cut-off: 60 dB (L_DEN)_ 55 dB (L_N_); <45 dB ≥45–<55 dB ≥55–<65 dB ≥65–<75 dB ≥75 continuous (starting at 0 dB)	Odds Ratio (adjusted) yes
van Kamp et al. 2007 [54]	NL longitudinal/ cross-sectional and panel study (+)	*n* = 6091 *n* = 2700 (longitudinal) ≥18 years	GHQ-12	aircraft	L_DEN_ L_N,10:00 p.m.-6:00 a.m._	-	Odds Ratio (adjusted) yes
Zock et al. 2018 [55]	NL cross-sectional (analysis within cohort) (+)	*n* = 4450 0–≥65 years	Primary care data ^i^	road railway	L_DEN_	-	Odds Ratio (adjusted) yes

4DSQ four-Dimensional Symptom-Questionnaire; BDI-II Beck Depression Inventory-II; CES-D Center for Epidemiological Studies Depression Scale; CIDI semi-structured Composite International Diagnostic Interview; CIS-R Clinical Interview Schedule-Revised; HADS-D Anxiety and Depression Scale; HSCL Hopkins Symptom Checklist; GHQ General Health Questionnaire; UK United Kingdom; NL Netherlands, NAT number above threshold; PHQ-9 nine-item Patient Health Questionnaire; K10 Kessler Psychological Distress Scale; F female; M male. (a) noise levels started at this level, and all noise levels below this range were set to this level; (b) Composite International Diagnostic Interview (CIDI) using DSM-IV criteria; (c) International Statistical Classification of Diseases and Related Health Problems version 9 ICD-9: 311 and ICD-10: F33, F34 diagnoses; (d) ICD-9 296.2, 296.3, 300.4, 309.28, 311; ICD-10 F32–F34.1, F41.2; (e) Anatomical Therapeutic Chemical (ATC) classification code: N06A prescription from National Prescription Register; (f) NAT_55_ = the number above threshold 55 dB, L_max55_ = average maximum sound level above threshold 55 dB; (g) ICD-10: F32.–, F34.1, F33.–, F41.2; (h) Continuous analysis considered noise levels starting at 35 dB (noise below 40 dB set to 35 dB).; (i) International Classification of Primary Care (ICPC) code: P03, P76 from Nivel primary care database.

**Table 4 ijerph-17-06175-t004:** Anxiety.

Study	Region(s) Study Design Quality Score (++,+,-)	Population	Outcome Assessment	Noise Source	Noise Measures	Noise Categories	Effect Estimates; Meta-Analysis (Yes, No with Reasons)
Bocquier et al. 2014 [61]	France, Retrospective cohort (-)	*n* = 190,617 (M = 87,975, F = 102,642) 18–64 years	Prescriptions of anxiolytics, benzodiazepin-derivates and analogues (ATC code) ^a^	road	L_N, 10:00 p.m.-6:00 a.m_.	<45 dB ≥45–<50 dB ≥50–<55 dB ≥55 dB	Odds Ratio (adjusted) yes
Floud et al. 2011 [38]	UK, NL, Sweden, Italy, Germany, Greece, cross-sectional (-)	*n* = 4861 (M = 2404, F = 2457) 45–70 Years	Use of anxiolytics (last 2 weeks, self-reported)	aircraft road	L_eq, 7:00 a.m.-11:00 p.m._ or L_eq, 6:00 a.m.-10:00 p.m._ L_eq, 11:00 p.m.-7:00 a.m._ or L_N, 10:00 p.m.-6:00 a.m._	LAeq,16h ^b^ 35–76 dB (aircraft) ^b^ 45–76 dB (road) LN ^b^ 30–70 dB (aircraft) ^b^ 45–70 dB (road)	Odds Ratio (adjusted) yes
Generaal et al. 2019a [39]	NL case-control (-)	*n* = 2980 (M = 1007, F = 1973) 18–65 years	Recent diagnosis (anxiety disorder based on CIDI ^c^)	Aircraft Road rail (combined)	L_DEN_	*-*	Odds Ratio (adjusted) no, exposure sources were not considered separately
Greiser et al. 2010 [41]	Germany case-control (-)	*n* = 511,742 with anxiety disorder: *n* = 2344 (M = 709, F = 1635) >39 years	Routine health insurance data ^d^	aircraft	L_N, 10:00 p.m.-6:00 a.m._ L_eq, 6:00 a.m.-10:00 p.m._ L_eq, 11:00 p.m.-1:00 a.m._ L_eq, 3:00 a.m.-5:00 a.m._ L_eq,24h_	Leq,24h ≥35 dB otherwise ≥40 dB	Odds Ratio (adjusted) no, main noise effects not reported without interaction terms
Halonen et al. 2013 [59]	Finland cross-sectional (-)	*n* = 15,611 (M = 3086, F = 12,525) 21–76 years	Anxiolytics use (ATC code)	road	L_DEN_	≤45 dB 45.1–50 dB 50.1–55 dB 55.1–60 dB >60 dB	Odds Ratio (adjusted) yes
Jonah et al. 1981 [62]	Canada cross-sectional (-)	*n* = 1150 (adults)	STAI	road	L_eq,24h_	45–75 dB	Correlation coefficient no
Klompmaker et al. 2019 [45]	NL cross-sectional (+)	*n* = 354,827 (M = 161,045, F = 193,782) ≥19 years	Prescriptions of anxiolytics (ATC code ^e^)	road railway	L_DEN_	*-*	Odds Ratio (adjusted) yes
Nivision & Endersen 1993 [63]	Norway, cross-sectional (-)	*n* = 82	STAI	road	L_eq,24h_ L_max_	*-*	Correlation coefficient no
Okokon et al. 2018	Finland cross-sectional (-)	*n* = 5860 (M = 2497, F = 3363) 55.0 years	Use of anxiolytics (a) 1 week, (b) 1–4 weeks, (c) 1–12 months, (d) over 1 year ago	road	L_DEN_	≤45 dB, 45.1–50 dB, 50.1–55 dB, 55.1–60 dB ≥ 60 dB	Odds Ratio (adjusted) yes
Stansfeld et al. 1993, 1996 [49,50]	UK cohort (+) cross-sectional (-)	*n* = 2398 (only men) 50–64 Years	GHQ-30	road	L_eq,6:00 a.m.-10:00 p.m._	51–55 dB 56–60 dB 61–65 dB 66–70 dB	Means/ Percent (adjusted) yes (unadjusted RRs derived from distribution in 1993 paper)
Zock et al. 2018 [55]	NL cross-sectional (analysis within cohort) (+)	*n* = 4450 0–≥ 65 years	Primary care data ^f^	road railway	L_DEN_	*-*	Odds Ratio (adjusted) -yes

CIDI Semi-structured Composite International Diagnostic Interview; GHQ General Health Questionnaire; UK: United Kingdom; NL Netherlands; NAT number above threshold; (a) ATC code: N05B, N05CD, N05CF from French National Health Insurance Fund (NHIF) database; (b) noise levels started at this level, and all noise levels below this range were set to this level; (c) Composite International Diagnostic Interview using DSM-IV criteria; (d) ICD-9: 300 and ICD-10: F40, F41 diagnoses; (e) ATC-code: N05B prescription during study; (f) ICPC code: P01, P74.

**Table 5 ijerph-17-06175-t005:** Dementia and Alzheimer’s disease.

Study	Region(s) Study Design Quality Score (++,+,-)	Population	Outcome Assessment	Noise Source	Noise Measures	Noise Categories	Effect Estimates; Meta-Analysis (Yes, No with Reasons)
Andersson et al. 2018 [64]	Sweden, Cohort (+)	*n* = 1721 (M = 985, F = 736) 55–85 years	Alzheimer’s disease and vascular dementia: Three-step procedure: general examination, examination by specialists and diagnosed by specialist ^a^	road	L_eq,24h_	<55 dB ≥55 dB	Odds Ratio (adjusted)
Carey et al. 2018 [65]	UK, Cohort (+)	*n* = 130,978 (M = 65 130, F = 65,848) 50–79 years	Alzheimer’s disease and vascular dementia: Doctor’s diagnosis (from database)	road	L_N, 11 p.m.-7:00 a.m._	Continuous analysis	Odds Ratio (adjusted)
Fuks et al. 2019 [67]	Germany, Cohort (+)	*n* = 288 (females only) 74.2 years (±2.2)	Cognitive function according to CERAD-Plus battery	road	L_DEN_ L_NIGHT_		Odds Ratio (adjusted)
Linares et al. 2017 [66]	Spain, longitudinal ecological time series study (–)	*n* = 3,116,897	Dementia-related emergency hospital admissions (related to organic psychoses: ICD-9 ^b^)	road	L_N, 10:00 p.m.-8:00 a.m._ L_eq, 8:00 a.m.-10:00 p.m._	Continuous analysis	Correlation coefficient no
Tzivian et al. 2016 [68]	Germany cross-sectional (analysis within cohort) (+)	*n* = 2050 50–80 years	Mild cognitive impairment, test battery with ADAS and NAI ^c^	road	L_DEN_ L_N,10:00 p.m.-6:00 a.m._	Dichotomized (cut point 60 dB (LDEN) and 55 dB (LN) <45 dB ≥45–<55 dB ≥55–<65 dB ≥65–<75 dB ≥75 Continuous	Odds Ratio (adjusted)

ADAS = Alzheimer’s disease Assessment Scale, CERAD = Consortium to Establish a Registry for Alzheimer’s Disease NAI = Nuremberg Gerontopsychological Inventory, UK = United Kingdom; (a) Mini-Mental State Examination (MMSE) score of ≤23 and medical records; diagnosis according to the DSM-IV (Diagnostic and Statistical Manual of Mental Disorders, fourth edition) and NINCDS–ADRDA (National Institute of Neurological and Communicative Diseases and Stroke/Alzheimer’s Disease and Related Disorders Association) criteria) Mini-Mental State Examination (MMSE) and total score used; (b) ICD-9 codes: 290.0–290.2, 290.4–290.9, 294.1–294; (c) Diagnosis according to the criteria of Petersen [69]: presence of a subjective cognitive complaint, or presence of an objective cognitive impairment that did not fulfill the criteria for dementia, and activities of daily living were generally intact.

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
