# Peer review of "Traffic Noise and Mental Health: A Systematic Review and Meta-Analysis"

_ijerph, 2020, doi:10.3390/ijerph17176175_

Round 1
Reviewer 1 Report
It is a good experience to read this review paper on the relationship between traffic-related noise exposure and mental health. The structure is well-organized and the methodology is scientifically sound. I also believe this systematic review contributes to better conclude the noise exposure-mental health association. However, there are still some problems mainly in the Introduction, Methods and Discussion sections:
Major problems:
- In Introduction and Discussion, 3 other review papers on the same topic are elaborated but it may not justify the reason for conducting a similar review again, especially compared with the latest work by Dzhambov and Lercher (2019). I think more contents should be supplemented in Introduction on why the demarcation of traffic noise (i.e., road noise, aircraft noise and railway noise) is necessary to understand its associated mental health problems.
- Again in Methods and Results, the majority of the reviewed studies are in low quality, especially for studies on aircraft noise and railway noise. This might underplay the results of meta-analysis, so I recommend more should be discussed on the problems existing in previous empirical studies and the ways to put forward in Discussion.
Minor issues:
- In L106 P3, it says “We included studies published in any language”, but in L121 on the same page the statement changes into “We used no geographic or language restrictions though a German or English title and abstract was an inclusion criteria”. Will this selection produce the language bias on overemphasizing the Germany research?
- L142-144 P4: Did the previous research have different confounders included because it may to some extent determine the modeling effect of traffic noise? It would be glad to discuss more on this issue.
- L160-161 P4: I am curious how to convert the noise exposure values to Lden based on the reported results of previous studies? Also I cannot refer to the Brink et al. (2018) in the reference list.
- L255-256 P10: It is ambiguous for the explanation of I2. What does the heterogeneity mean and how it is calculate based on several research findings?
- L273-275 P11, L317-318 P14 and L323 P16: The number of studies on aircraft noise and railway noise is highly limited. Is it the reason why previous reviews did not conduct a detailed division on sources of traffic noises or mainly paid attention to road traffic noise? This is important since this is the main argument as stated in the paper.
- L491-495 P18: It is great to discuss on the different attributes of traffic-related noise sources. Indeed, the occasional aircraft noise or railway noise may relate to mental health in a different way compared with the persistent road noise in the daytime?
- P507 L19: Why the noise annoyance appears in the conclusion? I would say annoyance is a mediating pathway between noise exposure and mental disorders (e.g. depression and anxiety), so the policy on reducing noise annoyance may have the same effect on preventing mental disorders.
- I strongly suggest to reorganize the Discussion section. Now the contents are redundant on the contrast with previous reviews of the same topic and limitations on measurement of mental health. It would be better to learn more on the limitations of previous empirical studies and some suggestions on future research.
Author Response
Reviewer 1
It is a good experience to read this review paper on the relationship between traffic-related noise exposure and mental health. The structure is well-organized and the methodology is scientifically sound. I also believe this systematic review contributes to better conclude the noise exposure-mental health association. However, there are still some problems mainly in the Introduction, Methods and Discussion sections:
Major problems:
- In Introduction and Discussion, 3 other review papers on the same topic are elaborated but it may not justify the reason for conducting a similar review again, especially compared with the latest work by Dzhambov and Lercher (2019). I think more contents should be supplemented in Introduction on why the demarcation of traffic noise (i.e., road noise, aircraft noise and railway noise) is necessary to understand its associated mental health problems.
Thank you for your kind comments and for this suggestion. We now include the following text in the introduction:
“In addition, there are indications that not just traffic noise intensity but the source of traffic noise might affect mental health. An emotional response to noise in the form of annoyance may be one way that noise contributes to mental health problems [7], and the WHO review on noise annoyance shows that noise annoyance varies greatly between traffic noise sources [8]. These reactions to noise could be due to the diverse characteristics of noise from each traffic source. Fluctuations in noise differ for each form of traffic noise, and the intermittency ratio (a metric of noise fluctuation) is positively associated with annoyance to railway and aircraft noise [9]. Traffic related sleep disturbances might also mediate the pathway between traffic noise exposure and mental health problems. Research demonstrates that each traffic noise source differentially affect subjective and objective sleep measures. Road and railway noise are more likely to cause an awakening than aircraft noise [10], while aircraft and railway noise negatively influence subjective sleep quality [11]. Our own study of traffic noise found aircraft traffic noise between >45 - ≤50 dB increased the odds of an incident depressive episode by 1.18 (95% CI 1.16–1.21) [12]. On the other hand, we observed a similar increase in odds for much higher road traffic noise of ≥70 dB (OR = 1.17; 95% CI 1.10–1.25). Therefore, it is important to differentiate between various sources of traffic noise when considering their associations with mental health problems.”
- Again in Methods and Results, the majority of the reviewed studies are in low quality, especially for studies on aircraft noise and railway noise. This might underplay the results of meta-analysis, so I recommend more should be discussed on the problems existing in previous empirical studies and the ways to put forward in Discussion.
We added a discussion of the problems of previous empirical studies and make suggestions for future research:
“However, most of the included studies in this systematic review were of low study quality. There is a need for high quality prospective population-based studies of railway noise, aircraft noise, and studies considering traffic noise exposure and dementia. All studies on traffic noise suffer from some degree of misclassification bias due to inexact assessments of noise exposure that, at best, are modelled to one spot on the outside of a home. Without information on how long study participants actually spend at home, there is likely to be both over- and underestimation of the true exposure. The common convention of using the loudest house façade may also systematically overestimate noise exposure, and better traffic noise measurements will improve the precision of future studies [84]. Improved noise measurements in future studies might also help attenuate the problem of heterogeneity, which was considerable in the meta-analyses for road and railway traffic noise. However, methodological diversity usually occurs in meta-analysis. Therefore, it is argued that heterogeneity is inevitable [20].”
Minor issues:
- In L106 P3, it says “We included studies published in any language”, but in L121 on the same page the statement changes into “We used no geographic or language restrictions though a German or English title and abstract was an inclusion criteria”. Will this selection produce the language bias on overemphasizing the Germany research?
We rephrased the sentence into: “We included studies published in any language with an English or German title and abstract”.
We believe that this selection has a very little language bias effect since most research applying to our predefined inclusion criteria is published in English in the used electronic databases and conference contributions.
- L142-144 P4: Did the previous research have different confounders included because it may to some extent determine the modeling effect of traffic noise? It would be glad to discuss more on this issue.
Yes, previous research included different confounders. Included confounders are presented in the “Comments” section of the detailed extraction form in the supplementary section. Confounding was evaluated according to SIGN and CASP hybrid tool for study quality rating. We included a description in the text:
“For example, the tool includes questions on i.e. appropriate recruitment of study population, measurement of exposure and outcome. Also, it is evaluated if all important confounding factors (in our study age, sex and education/ socioeconomic state) have been considered and taken into account in the design and analysis.”
- L160-161 P4: I am curious how to convert the noise exposure values to Lden based on the reported results of previous studies? Also I cannot refer to the Brink et al. (2018) in the reference list.
Thank you for bringing this to our attention. We included the formulas (now Table 2) and the reference in the text.
- L255-256 P10: It is ambiguous for the explanation of I2. What does the heterogeneity mean and how it is calculate based on several research findings?
Ideally, studies included in the meta-analysis should all be performed in a similar way (study design, similar study protocol). Heterogeneity of included studies describes the amount to which this ideal is not met. In this study, we used the measure I2 that describes the percentage of variance that is attributable to study heterogeneity and not due to chance alone. If each included study could be considered to have repeated the same study on a similar population with the same methods (i.e. repetition of the same study), the observed differences in results are result of chance alone. The I2 statistic quantifies the amount of variance not due to chance with a metric that can be compared to other meta-analyses. We included the information: “Heterogeneity was quantified by calculating I2 according to the following formula: I2 = (Q – df/Q) x 100 where Q is the chi-squared statistic and df is the degrees of freedom (number of studies – 1) [19,20]. The Q chi-squared statistic is the weighted sum of the differences between each study's effect estimate and the pooled estimate squared.”
- L273-275 P11, L317-318 P14 and L323 P16: The number of studies on aircraft noise and railway noise is highly limited. Is it the reason why previous reviews did not conduct a detailed division on sources of traffic noises or mainly paid attention to road traffic noise? This is important since this is the main argument as stated in the paper.
Yes, for anxiety disorders the number of studies was limited for railway and aircraft noise. Dzhambov & Lercher (2019) chose to only focussed on road traffic noise using a meta-analytic approach. Clark et al. 2020 also included results on railway noise and aircraft noise but did not summarized results meta-analytically. We think the new paragraph in the introduction helps illustrate our motivation for looking at all three forms of traffic noise.
- L491-495 P18: It is great to discuss on the different attributes of traffic-related noise sources. Indeed, the occasional aircraft noise or railway noise may relate to mental health in a different way compared with the persistent road noise in the daytime?
Thank you, this is correct, noise fluctuations and maximum noise levels may exert stronger effects on depression risk than average noise levels. In the NORAH study, individuals with low average noise levels (below 40 dB) but with nightly maximum levels over 50 dB had an increased depression risk from aircraft noise. We included this information in the discussion (Lines 590-592).
- P507 L19: Why the noise annoyance appears in the conclusion? I would say annoyance is a mediating pathway between noise exposure and mental disorders (e.g. depression and anxiety), so the policy on reducing noise annoyance may have the same effect on preventing mental disorders.
Good point. The reasoning behind this conclusion was the fact that annoyance is probably mediating the pathway between noise exposure and mental health. So, while we cannot be sure that reducing traffic noise will prevent mental disorders, it should at least reduce annoyance. However, because annoyance is not the subject of our research, we excluded the noise annoyance from the conclusion and rephrased the sentence: “Public policies should reduce environmental traffic noise to prevent depression and anxiety disorders.”
- I strongly suggest to reorganize the Discussion section. Now the contents are redundant on the contrast with previous reviews of the same topic and limitations on measurement of mental health. It would be better to learn more on the limitations of previous empirical studies and some suggestions on future research.
Thank you for this suggestion. We added a paragraph to the discussion (see above).
Reviewer 2 Report
The submitted paper is of very high quality and only requires small changes. Since not only Stata but also other statistical programs are used for the analysis, a brief description of the content of the analysis steps carried out in these packages would be desirable instead of or in addition to naming the packages glst and metan. The naming of program packages is part but does not correspond to a complete method description. At the moment, the reader would have to look for the software and program packages in order to understand the analysis. Exact p-values should also be reported if significant results are mentioned. A clearer conclusion would also be desirable. Currently, this is a very short summary, in which no position is taken and the reader is not supported in drawing a conclusion or deriving next steps. Against this background, the reader could come to the conclusion that noise is not so bad at all, which may be a possible conclusion, but which should be weighed against the still too thin database, for example.
Author Response
Reviewer 2
The submitted paper is of very high quality and only requires small changes. Since not only Stata but also other statistical programs are used for the analysis, a brief description of the content of the analysis steps carried out in these packages would be desirable instead of or in addition to naming the packages glst and metan. The naming of program packages is part but does not correspond to a complete method description. At the moment, the reader would have to look for the software and program packages in order to understand the analysis. Exact p-values should also be reported if significant results are mentioned. A clearer conclusion would also be desirable. Currently, this is a very short summary, in which no position is taken and the reader is not supported in drawing a conclusion or deriving next steps. Against this background, the reader could come to the conclusion that noise is not so bad at all, which may be a possible conclusion, but which should be weighed against the still too thin database, for example.
- Thank you for your review and your kind comments. We have added some details to our elaborate on the use of the generalized least-squares models and random-effects meta-analysis:
“
- If risk estimates were reported for categories of noise, study-specific risk estimates per 10 dB linear increase of traffic noise were estimated by applying the generalized least-squares model for trend estimation of summarized dose-response data using the glst Stata package [10]. The generalized least-squares for trend estimation takes into account the fact that risk estimates from a single study do not fulfill the assumption of independence required of weighted linear regression. A generalized least-squares model was estimated for each applicable study separately as a fixed-effect model using the logarithm of the risk estimates as the dependent variable and the average of the noise level categories (with the reference group set to zero) as the independent variable. If necessary, noise exposure values were converted to day-evening-night weighted-24 hour means (LDEN) according to Brink, et al. [14] prior to the modelling.
Table 2. Converting to LDEN according to Brink, et al. [14]
Converting |
Road traffic noise |
Railway traffic noise |
Aircraft noise |
Leq24h to LDEN
|
Leq24h + 3.3 dB |
Leq24h + 6.1 dB |
Leq24h + 3.6 dB |
LD to LDEN
|
LD + 2.0 dB Leq, 16h + 1.5 dB |
LD + 6.0 dB Leq, 16h + 5.9 dB |
LD + 2.1 dB Leq, 16h + 2.3 dB |
LDN to LDEN
|
LDN + 0.7 dB |
LDN + 0.4 dB |
LDN + 1.1 dB |
LN to LDEN
|
LN + 8.0 dB |
LN + 6,6 dB |
LN + 9.9 dB |
- The reported and self-calculated risk estimates per 10 dB Lden were pooled using a DerSimonian and Laird random-effects meta-analysis. A random-effects model was chosen, because heterogeneity between study populations can be expected. This method weights each effect estimate by the inverse of its (within-study) variation and the heterogeneity between studies (between-study variation). The Stata package metan was used to conduct the random-effects meta-analysis and to create forest plots [11]. "
- Regarding exact p-values, we intentionally refrained from reporting p-values, as their use has previously been discouraged by epidemiological journals (Lang et al. 1998). We prefer to indicate statistical significance by reporting 95% confidence intervals, because confidence intervals also provide an indication of the precision of estimates.
Lang JM, Rothman KJ, Cann CI. That confounded. P-value. Epidemiology 1998; 9: 7–8.
- We also amended our conclusions. As you state, the “database is thin”, so making any further recommendations may over-extrapolate our results, although we do find overall there are indications that traffic noise increases the risk for mental health problems. We now also mention the need for more quality research:
“Our results also suggest that road traffic noise may exert comparatively small effects on depression and anxiety risks, however few studies had an acceptable methodological quality and the study design (i.e. outcome and exposure assessment) varied greatly. We found too few studies to make conclusions on the effects of railway noise on mental health. We encourage future research to design high quality (prospective) studies assessing all traffic noise sources in the same population. In the meantime, public policies that reduce environmental traffic noise can help to prevent depression and anxiety disorders.”